# The Spatial Difference of "Internet plus Tourism" in Promoting Economic Growth

**Rijia Ding and Meng Huang ***

School of Management, China University of Mining and Technology (Beijing), Beijing 100083, China; dingrijia@cumtb.edu.cn
* Correspondence: bqt1800503024@student.cumtb.edu.cn

**Abstract:** The "Internet plus Tourism" mode, which is the coordination of the internet and tourism, has become a new driving force for regional economic growth. In order to investigate the mechanism and superiority of "Internet plus Tourism" in terms of economic growth compared with the independent effects of the internet and tourism on economic growth, this paper uses the DEA model to calculate the tourism efficiency of 30 provinces in China from 2011 to 2019 and three spatial econometric models to comparatively analyze the independent effects of the internet and tourism with the synergized effect of "Internet plus Tourism" on economic growth. The results show that (1) the overall pattern of the internet is that coastal areas are ranked higher; (2) tourism efficiency presents a polarized hierarchical structure; (3) The effect of "Internet plus Tourism" on economic growth is significantly positive and is significantly greater than the independent effects of the internet and tourism on economic growth; (4) the internet, tourism, and "Internet plus Tourism" have different effects on economic growth in different regions. Therefore, the paper suggests that China should accelerate the integration of "Internet plus Tourism" and realize sustainable economic development.

**Keywords:** internet; tourism development; "Internet plus Tourism"; economic growth; spatial metrology





## 1. Introduction

Tourism is the fastest value-added industry in China's national economy and has long been highly valued by the country. It not only plays an important role in driving regional economic growth but also strengthens the national economy by driving other related industries. COVID has caused heavy losses to the global tourism industry. How to maintain the vitality of the tourism market and maintain the survival and development of tourism under an epidemic situation has become an important issue in the field of tourism. As an emerging industry, the internet has taken a unique development path, driving the coordinated development of traditional industries and opening up new possibilities for traditional industries. In recent years, under the dual influence of policy guidance and market development, China's tourism industry has developed rapidly. The continuous broadening of the consumer market and the further expansion and optimization of tourism products are also gradually affecting the market layout. The industry is becoming increasingly diversified. In terms of obtaining travel information, traditional travel agencies, large-scale online travel platforms, and consumer original content are still the three most common channels that consumers consult. Among them, online tourism has developed by leaps and bounds in recent years, which has had a huge impact on the tourism industry and signaled a new era of transformation. It not only increases the diversity of tourism supply, provides the tourists with a diversity of tourism consumption behavior, and improves the tourists' willingness to revisit, but also reduces the waste of tourism resources and realizes sustainable tourism development through the rational allocation of tourism factor endowment. Although the internet has achieved great-leap-forward development and has rapidly changed the behavior of Chinese tourists, there is still a lingering question surrounding the impact of "Internet plus Tourism" on regional economic growth.

At present, scholars generally believe that the internet can significantly promote economic growth. They argue that technological innovation brought about by the internet will bring new changes to production systems, accelerate business innovation, and enable enterprises to maintain their competitiveness in an increasingly turbulent market [1,2]. At the same time, it promotes the deep integration of the internet and the real economy, which offers increasing returns and stronger innovation capabilities [3,4]. Jiménez et al. [5] and Salahuddin et al. [6] have confirmed the above viewpoints through empirical studies. Tourism economists have conducted extensive research on the relationship between tourism development and regional economic growth. Generally speaking, the relevant research has formed four different viewpoints. The first viewpoint argues that tourism development can promote economic growth, leading to "tourism-led economic growth" (TLEG) [7–9]. The second argues that tourism development and economic growth are mutually causal, as demonstrated via the "reciprocal economic growth hypothesis" (RCGH) [10]. The third argues that tourism development benefits from economic growth, forming "economic-driven tourism growth" (EDTG) [11–13]. The fourth and final argument states there is no relationship between tourism development and economic growth and claims it is an "irrelevant hypothesis" (IH) [14,15]. Among them, the majority of the literature supports the first conclusion. In recent years, research on the relationship between the internet and tourism has also gradually increased, and the relevant literature has basically reached a consensus on the important contribution of the internet to tourism development [16]. It is generally believed that the internet enhances the strategic position of the industry and offers it a competitive advantage. It plays an important role in shaping the image of destinations [17]. In addition, researchers have focused on the application of the internet in the tourism industry [18] and the online paths and mechanisms used to promote tourism [19,20], believing that internet developments enable the tourism industry to surpass the constraints of geographic space, drive the modular transformation of tourism enterprises, build new cooperative operation mechanisms, and meet the diverse needs of tourists [21]. There are also a large number of studies focusing on the field of micro-enterprise management, specifically analyzing the application model of the internet in tourism enterprises [22].

However, most studies have studied the relationship between two of the categories of the internet, tourism development, and economic growth; few studies have integrated the three into the same research framework, and the mechanism by which "Internet plus Tourism" coordinates the development of the internet and tourism and affects economic growth is unclear. Thus, this paper creatively divides the impact of the internet and tourism development on economic growth into an independent effect and a synergistic effect. First, in the "theoretical mechanisms" section, we analyze the theoretical basis and establish the theoretical model of tourism development, the internet, and "Internet plus Tourism" to promote economic growth. Secondly, in the "research methods and data sources" section, we explain the data sources of the internet index, tourism efficiency, economic growth, and control variables. Then, the two methods of the data envelopment analysis (DEA) and spatial econometric mode are briefly introduced. In the section on empirical results and analysis, we use the DEA model to measure tourism efficiency and then use the spatial econometrics method, from the perspectives of whole and regional space to comparatively analyze the independent effects of the internet and tourism with the synergized effect of "Internet plus Tourism" on economic growth. Finally, the conclusions and suggestions of this article are given. The research has not only theoretical value but also practical significance for the promotion of the high-quality development of tourism and the promotion of China's economic growth.

## 2. Theoretical Mechanisms

### 2.1. Tourism Development Promotes Economic Growth

Foreign research on tourism development promoting economic growth began in the late 19th century and flourished in the 1960s. The earliest research on tourism and the economy was a statistical consideration of Bodio, Italy [23]. To extend the breadth of

research, we need to determine the specific driving mechanisms of tourism that promote economic growth. Theoretically speaking, these mechanisms can be divided into three aspects. The first is a spillover effect. In the theory of economic externality, spillover refers specifically to the externality of economic activities. According to new economic growth theory, the spillover effect is an important dynamic mechanism for explaining economic growth; as a spillover of the export trade, tourism can increase foreign exchange income and local wealth, boost material capital accumulation, attract investment by improving the investment environment, and increase the efficiency of destination-related enterprises by developing competition. The above ways will promote the improvement of comprehensive input-output efficiency [24,25]. The second aspect is the scale economy. Marshall's theory of the scale economy reflects the economic effects of large-scale production. Marx holds that the productivity of social labor must be based on large-scale production and cooperation. The development of tourism can expand actual market demand, reduce long-term average production costs, promote economies of scale, and improve comprehensive output efficiency [26,27]. The third aspect is the population capital effect; that is, tourism promotes the improvement of comprehensive efficiency by improving the level of human capital. The human capital theory centers on the economic value of humanity and its important role in the development of the modern economy and society. As a labor-intensive industry, tourism absorbs a large number of low-tech or even unskilled personnel. The labor force will receive relevant professional training through employment, which will play an important role in improving labor skills, and a strengthened, technical labor force will further contribute to economic efficiency [28].

**Hypothesis 1:** *Tourism can promote economic growth through its spillover effect, scale economy, and population capital effect.*

*2.2. The Internet Promotes Economic Growth*

With the advancement in theory and the need for market development, economic growth mechanisms have experienced a shift from capital, labor, and natural resources toward investment and technological innovation. Schumpeter first discussed how technological innovation promotes economic growth. Following Schumpeter's concept of innovation diffusion, scholars then put forward the theory of the "learning by doing" effect and technology spillover to explain the dynamic mechanism of technological innovation and its growth effects. The internet is a typical representative of information and communication technology and technological innovation. Research on the economic effects of the internet is subordinate to higher-level research on the impact of broader technological innovation. The impact of the internet and other information technologies on economic growth can be roughly divided into "technology effects" and "information effects". Technology effects mainly refer to the ways the internet can rapidly accelerate the technological progress and production efficiency of production departments [1], making the price of internet products fall and promoting internet users' capital accumulation. Through the substitution of internet capital for other capital, society as a whole can realize capital deepening [29], which then promotes economic growth. Information effects mainly refer to the impact of internet use on personal professional choices and market information acquisition. The "technology bias theory" proposed by Mincer [30] points out that high skilled workers can use new technology more efficiently to improve their labor productivity. Moreover, Xu et al. [31] found that the continual supply of relevant information services can enable people to cross the "digital divide" and enjoy "information dividends". Dettling [32] found that the use of the internet provides job seekers with wider access to employment information, improving workplace efficiency and reducing frictional unemployment. In addition, recruitment information publishers can use internet platforms to expand the scope of information transmission, reduce the cost of obtaining information, and, thus, reduce job vacancy times [33].

**Hypothesis 2:** *The internet can promote economic growth through its "technology effects" and "information effects".*

*2.3. "Internet plus Tourism" Promotes Economic Growth*

Scholars generally believe that the internet promotes the development of tourism in two ways: one is the ability of the internet to meet the growing needs of broad masses of tourists and improve consumer satisfaction. Tourists can independently choose and combine elements of tourism with the help of online platforms [34,35]. The tourism industry has evolved into an "information-driven" customized multi-supply system based on tourists' diverse demands, which operates according to "reactive supply" [34]. What is more, the penetration of the internet reduces the industry's operation costs, strengthens its sales ability, and greatly improves operation efficiency [36]. The internet allows travel agencies, hotels, scenic spots, and other tourism enterprises to revitalize their marketing methods and organizational performance. Wechat and other online platforms have improved the precision of marketing, stimulated market demand, increased customer stickiness, and greatly enhanced production efficiency [37].

Tourism development and internet technology do not exist independently. The integration of the two allows for the spread of technical features from other industries in relatively advantageous positions into various activities of the tourism industry, thereby expanding the industry's related channels and development potential. In taking internet technology as the core variable with which to study the spillover effect of tourism development on economic growth, we also need to study the contribution of the interaction and integration of tourism development and internet technology to economic growth. Therefore, based on the above analysis, as well as considering integration development theory, connection effect theory, and spillover effect theory, we also must examine the integration correlation effect of these two areas. This interaction promotes economic growth through coupling interaction spillover and integration correlation spillover. A mechanism model displaying how tourism development and internet technology promote economic growth is shown in Figure 1. Tourism development promotes economic growth through a tourism spillover effect, internet technology promotes economic growth through technology spillover, and the interaction between tourism development and internet technology promotes economic growth through coupling interaction spillover and integration correlation spillover.

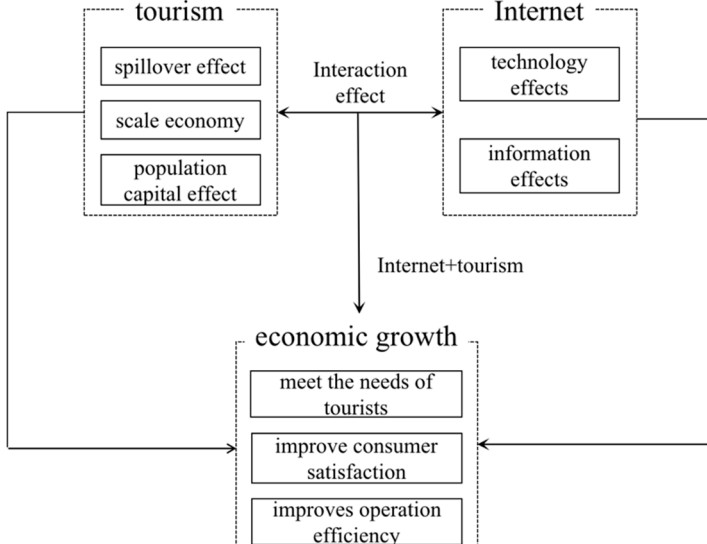

**Figure 1.** The oretical framework for the internet, tourism, and economic growth.

**Hypothesis 3:** *"Internet plus Tourism" has a positive effect on economic growth through meeting the needs of tourists and improving consumer satisfaction and operation efficiency.*

## 3. Research Methods and Data Sources

### 3.1. Data Sources

#### 3.1.1. Explanatory Variable—The Internet Index (Internet)

At present, most of the research papers on the internet have considered using the amount invested by enterprises on information processing equipment or the internet popularization rate to express internet development. In order to more accurately explain internet infrastructure and public utilization levels, this article uses the internet index to represent the development of the internet. The internet index can be calculated through a logarithm of the product of levels of internet penetration and the length of internet optical cable lines [38]. It does so for a number of reasons. First, these measurements reflect the degree of social utilization of the internet and the level of internet infrastructure. Second, product interaction represents the multiplier effect of the internet. Third, considering that the internet is measured from the perspective of integration, there will be an endogenous problem among variables. It is more appropriate to quantify it from the perspective of society as a whole to ensure the exogeneity of the variables. Fourth, it is difficult to build an index system to quantify because there are few related indicators of the internet in the data. In order to ensure the authenticity of its results, this paper uses panel data from 30 provinces and autonomous regions in China from 2011 to 2019 as the research object. The data are all from the China Statistical Yearbook.

#### 3.1.2. Explanatory Variable—Tourism Efficiency (Tourism)

The paper uses tourism efficiency to represent the development of tourism. Due to the lack of statistical data regarding land input indicators in tourism research, few scholars have included land elements in input index systems. However, the number of star hotels and scenic spots make up for the lack of land input elements to a certain extent. This paper also does not include land elements in its input index system. The labor input index reflects the number of tourism practitioners. The capital investment index mainly covers tourism services and tourism infrastructure. This paper uses the number of travel agencies, hotels with stars, and A-level scenic spots to measure capital investment. Therefore, the initial input indicators of this paper are the number of tourism practitioners, the number of travel agencies, the number of hotels with stars, and the number of A-level tourist attractions (spots), and tourism output is mainly represented by the number of tourists and tourism income. The input–output data used in the evaluation of tourism efficiency mainly comes from the China Tourism Statistical Yearbook and the China Statistical Yearbook.

#### 3.1.3. Response Variable—Economic Growth (Economy)

This paper uses the logarithm of per capita GDP to measure economic development. GDP per capita considers the actual GDP per capita of each region. Actual GDP per capita is obtained by deflating the nominal GDP per capita of each province and city in the China Statistical Yearbook using the same year's month-on-month growth index (100 in the previous year) [39].

#### 3.1.4. Control Variables

Transportation infrastructure (transportation): The availability and accessibility of transportation infrastructure affect the spatial directions of tourism flow, tourism demand, and tourist behavior [40], thus affecting the local economy. In this paper, a logarithm of the total mileage of railways, highways, and water transportation in each region is used to measure the degree of traffic infrastructure in each province.

Industrial structure (industrial): Industrial structure creates and provides the necessary factors for the development of tourism that not only support the scale development of tourism but also offer an important guarantee to enhance its economic contribution.

Therefore, this paper uses the proportion of GDP output of the tertiary industry to express industrial structure.

Openness (openness): The proportion of total import and export within the GDP reflects the degree of each region's openness and can be used to evaluate the economic growth effect of foreign tourists attracted by the openness of provinces on the efficiency of local tourism.

Due to the large values of the internet, economic growth, and transportation infrastructure indicators and the inconsistency of three indicators' measurement units, this paper adopts a logarithmic form for these three indicators to reduce the absolute difference between the data and to eliminate heteroscedasticity. Other indicators are in the form of ratios, which do not need to be logarithmic.

The indicators used in this paper are described in Table 1.

**Table 1.** Index system.

| Index | Specific Indicator | | Calculation Formula |
|---|---|---|---|
| Internet index | level of internet penetration | | level of internet penetration × length of internet optical cable lines |
| | length of internet optical cable lines | | |
| Tourism efficiency | Input index | the number of tourism practitioners | DEA |
| | | the number of travel agencies | |
| | | the number of hotels with stars | |
| | | the number of A-level tourist attractions | |
| | Output index | the number of tourists | |
| | | tourism income | |
| Economy | per capita GDP | | - - - - - - - - - |
| Transportation | total mileage of railways | | Log(mileage of railways+ mileage of highways+ mileage of water transportation) |
| | total mileage of highways | | |
| | total mileage of water transportation | | |
| Industrial | output value of the tertiary industry | | output value of the tertiary industry/GDP |
| | gross domestic product (GDP) | | |
| Openness | total import and export | | total import and export/GDP |
| | gross domestic product (GDP) | | |

### 3.2. Research Methods

#### 3.2.1. Data Envelopment Analysis

DEA is the most common nonparametric method for evaluating the relative efficiency of input and output. The study compared the relative efficiency and benefits of several decision-making units (DMUs) with multiple inputs and outputs. The DMU refers to the units that measure performance, expressed by K (k = 1, 2, . . . , n). The basic idea is to find the minimum cone of all DMU production sets, and the boundary of the convex cone is the best production front. The production status of tourism in the studied regions is compared with this frontier, and, finally, the relative measurement results of their efficiency are obtained. DEA has an absolute advantage in dealing with complex multi-output and multi-input problems. L stands for input indicators and M for output indicators. $x_{jl}$

represents the input of $l$ resource of unit $j$. $y_{jm}$ represents the output of m of unit $j$. The DEA model is expressed as follows:

$$
\begin{cases}
min\left(\theta - \varepsilon\left(e_1^T s^- + e_2^T s^+\right)\right) \\
s.t. \sum\limits_{j=1}^{k} x_{jl}\lambda_j + s^- = \theta x_l^n, \quad l = 1, 2, \ldots, L \\
\sum\limits_{j=1}^{k} y_{im}\lambda_j - s^+ = y_m^n, \qquad m = 1, 2, \ldots, M \\
\lambda \geq 0, \qquad\qquad\qquad n = 1, 2, \ldots, K
\end{cases}
\tag{1}
$$

Given a set of input and output $(x_{jl}, y_{jm})$, $\theta$ is the overall efficiency; $\lambda_j$ is the weight variable; $s^-$ and $s^+$ are the slack variable and the residual variable, respectively; $\varepsilon$ is the non-Archimedean infinitesimal quantity; $e_1^T = (1, 1, \ldots, 1) \in E_m$ and $e_2^T = (1, 1, \ldots, 1) \in E_k$ are m-dimensional and k-dimensional unit vector spaces, respectively.

3.2.2. Spatial Econometric Model

(1) Spatial correlation test

Before the introduction of the spatial econometric model, it is necessary to measure the spatial dependence of tourism and economic growth in China. This paper uses the global Moran's I index and its statistical test to analyze the data's spatial dependence characteristics.

$$
Moran's\ I = \frac{\sum\limits_{i=1}^{n}\sum\limits_{j=1}^{n} W_{ij}\left(X_i - \overline{X}\right)\left(X_j - \overline{X}\right)}{\frac{1}{n}\sum\limits_{i=1}^{n}\left(X_i - \overline{X}\right)^2 \sum\limits_{i=1}^{n}\sum\limits_{j=1}^{n} W_{ij}}
\tag{2}
$$

Moran's I is generally in the range of $[-1, 1]$. When it is less than 0, it means that the space is negatively correlated. When it is equal to 0, it means that the space is not correlated. When it is greater than 0, it means that the space is positively correlated. Where $W_{ij}$ is the spatial weight matrix, $\overline{X} = \frac{1}{n}\sum_{i=1}^{j} X_i$ denotes the spatial proximity of the n regions. The commonly used 0–1 adjacency matrix is set as follows:

$$
W_{ij} = \begin{cases} 1 & \text{When region } i \text{ adjacent to } j. \\ 0 & \text{When region } i \text{ is not adjacent to } j \end{cases}
$$

(2) Model selection and construction

The spatial correlation test determined that tourism and economic growth have a spatial autocorrelation. Then, a spatial econometric model was used to study the impact of the internet and tourism development on economic growth. The commonly used spatial econometric models are the spatial autoregressive model, the spatial error model, and the spatial Durbin model.

The spatial autoregression model (SAR) adds the lag term of the explained variable to the classical regression model. It is generally used to study the influence of the behavior of adjacent areas on other areas within a region.

$$
economy_{it} = c + \rho \times Weconomy_{it} + \alpha_1 \times Internet_{it} + \alpha_2 \times tourism_{it} + \alpha_3 \times X_{it} + \varepsilon_{it} \tag{3}
$$

The spatial error model (SEM) is based on the classical regression model and considers the spatial disturbance error term, which means the influence of changes to the explanatory variables in adjacent areas on the response variable.

$$
economy_{it} = c + \beta_1 \times Internet_{it} + \beta_2 \times tourism_{it} + \beta_3 \times X_{it} + \varepsilon_{it} \tag{4}
$$

The spatial Durbin model (SDM) comprehensively considers the spatial lag factors of explanatory variables and explained variables. This study uses the spatial Durbin model given by Anselin and Lesage for reference and sets the model as follows:

$$economy_{it} = c + \rho \times Weconomy_{it} + \gamma_1 \times Internet_{it} + \gamma_2 \times tourism_{it} + \gamma_3 \times X_{it}$$
$$+\theta_1 \times WInternet_{it} + \theta_2 \times Wtourism_{it} + \theta_3 \times WX_{it} + \varepsilon_{it} \tag{5}$$

In the formula, $economy_{it}$, $Internet_{it}$, and $tourism_{it}$ represent economic growth, the internet, and tourism development in region i in the year t. $X_{it}$ represents the control variable, $W$ represents the spatial-weight matrix, and $\varepsilon_{it}$ represents a random disturbance term.

## 4. Empirical Results and Analysis

### 4.1. Internet Index Analysis

The internet index was devised via a logarithm of the internet penetration rate of 30 provinces in China from 2011 to 2019 and the length of internet optical cable lines. Using the natural breakpoint method in ArcGIS software, the internet index of 30 provinces was divided into 5 levels (Figures 2 and 3).

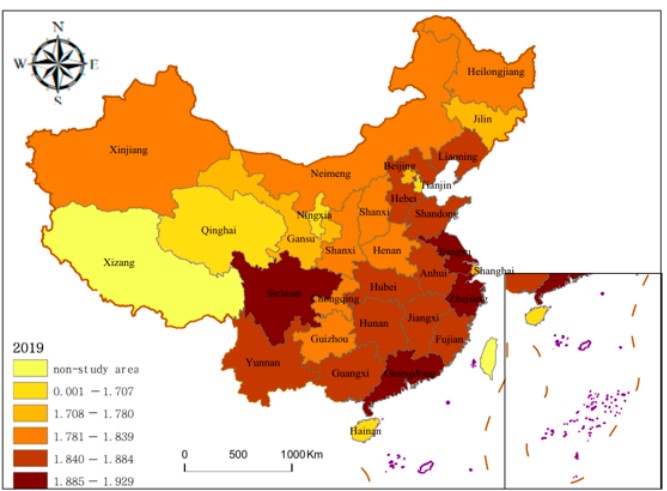

**Figure 2.** Spatial pattern of China's internet index in 2011.

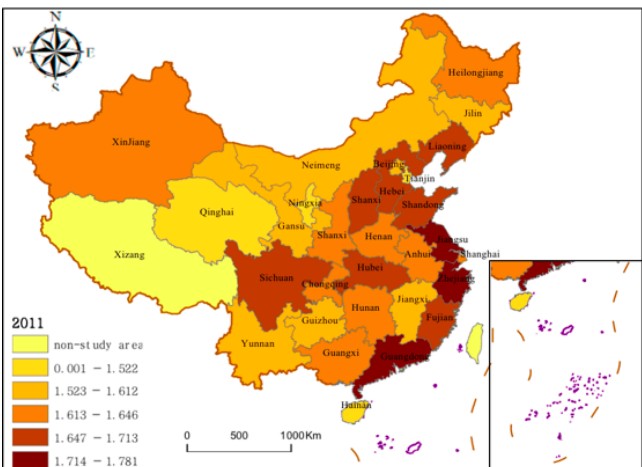

**Figure 3.** Spatial pattern of China's internet index in 2019.

(1) There are significant differences in the internet index across the provinces, and the index points migrate over time. In 2011, the proportion of provinces ranked from the first to fifth levels of the index were 36.67%, 23.33%, 23.33%, 13.33%, and 3.34%. The number of regions below the third level accounted for 80%, indicating that the internet index was

generally more in the middle and lower levels and the number of high-index areas was relatively small. In 2019, the overall internet index improved and the order of provinces also changed greatly. The number of areas below the third level accounted for only 63.33%.

(2) Regarding spatial distribution, the plate pattern is higher in coastal areas and lower in central and western inland areas. In 2011, the areas that ranked high on the internet index were mainly distributed across coastal areas, while central and western regions experienced a certain degree of the "collapse" phenomenon. By 2019, the central regions had improved. The growth of Shanxi, Jilin, Guizhou, and Guangxi was most significant, with prominent advantages in internet development. Internet resources in northwest China are scarce and restricted by a lack of transportation, difficult terrain, and other aspects, and their technological and digital disadvantages are becoming increasingly significant. They rely on the provinces higher in the index to diffuse their advances outward. On the whole, the eastern provinces have obvious advantages in terms of internet development, while central and western provinces struggle in this area; national internet development still needs to be further improved.

### 4.2. Tourism Index Analysis

Deap 2.1 software was used to calculate the efficiency of tourism (comprehensive efficiency) within 30 provinces in China from 2011 to 2019 (Table 2). The natural breakpoint method was used to divide them into five levels. The results of comprehensive efficiency measurements from 2011 to 2019 are presented using ArcGIS in Figures 4 and 5.

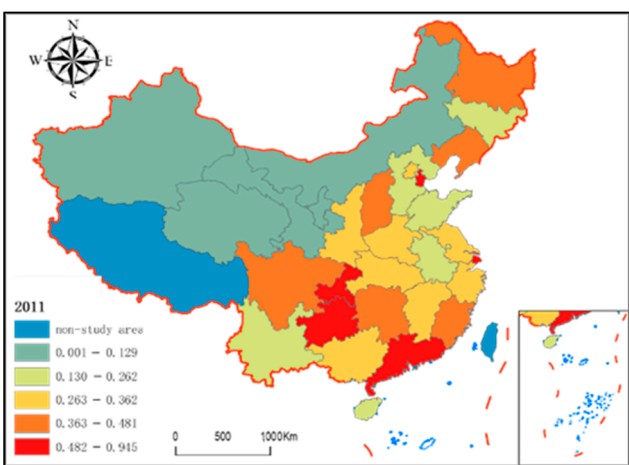

**Figure 4.** Spatial pattern of China's tourism distribution superiority in 2011.

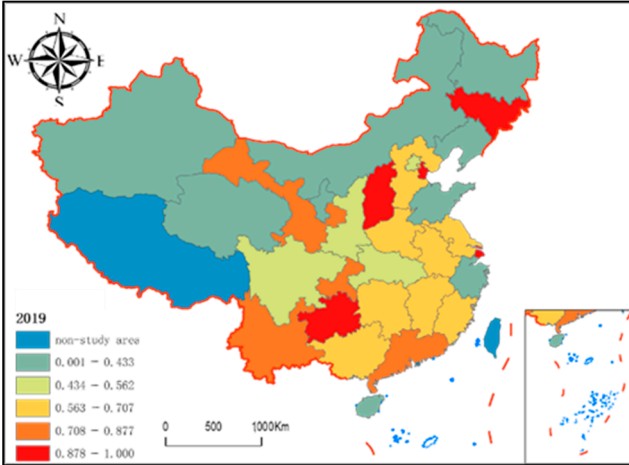

**Figure 5.** Spatial pattern of China's tourism distribution superiority in 2019.

(1) On the whole, the comprehensive efficiency of tourism shows an upward trend, and the average efficiency value rose from 0.346 in 2011 to 0.621 in 2019. In 2011, the comprehensive efficiency of Shanghai's tourism reached a good level, with an efficiency level of above 0.8. The city generated considerable tourism income and a high flow of visitors with low tourism factor input, and the comprehensive level of resource utilization was relatively high. Guizhou, Tianjin, Guangdong, and Chongqing were all in the middle level, with efficiency levels between 0.5 and 0.8; 83% of the other provinces had a comprehensive efficiency of less than 0.5, which signals a low efficiency of tourism development. By 2019, Tianjin, Shanxi, Jilin, Shanghai, Guizhou, Chongqing, Yunnan, and Gansu reached a good level, and four of them achieved the best possible efficiency, meaning their comprehensive efficiency was 1. The number of provinces with comprehensive efficiencies higher than 0.5 also increased to nine, indicating that the allocation and utilization level of tourism resources in each province changed from medium to good.

(2) Overall, the number of provinces with high-level tourism comprehensive efficiency increased. The proportions of provinces within the first level to the fifth level changed from 13.33%, 23.33%, 26.67%, 20.00%, and 16.67% to 16.67%, 13.33%, 26.67%, 13.33%, and 30%, respectively. However, the efficiency structure still presents a large number in the middle level and smaller numbers in the other levels. Some provinces have optimized resource allocation, and tourism development is good. For example, in Jiangxi, Guizhou, and Guangxi, tourism income increased by 773.09%, 761.84%, and 760.59%, respectively, during the study period, and the number of tourists increased by 395.8%, 566.37%, and 399.00%. Some efficiency levels remained stagnant. For example, in Liaoning, Heilongjiang, and other regions, the growth rate of output indicators was relatively low.

**Table 2.** Classification of tourism efficiency in 2011 and 2019.

| Level | 2011 | | 2019 | |
|---|---|---|---|---|
| | **Province** | **Proportion** | **Province** | **Proportion** |
| First level | Tianjin, Shanghai, Guangdong, Guizhou | 13.33% | Tianjin, Shanghai, Jilin, Shanghai, Guizhou | 16.67% |
| Second level | Shanxi, Liaoning, Heilongjiang, Fujian, Hunan, Chongqing, Sichuan | 23.33% | Guangdong, Chongqing, Yunnan, Gansu | 13.33% |
| Third level | Beijing, Jiangsu, Zhejiang, Jiangxi, Henan, Hubei, Guangxi, Shanxi | 26.67% | Hebei, Jiangsu, Anhui, Fujian, Jiangxi, Henan, Hubei, Guangxi | 26.67% |
| Fourth level | Hebei, Jilin, Anhui, Shandong, Hainan, Yunnan | 20.00% | Beijing, Hubei, Sichuan, Shanxi | 13.33% |
| Fifth level | Neimeng, Gansu, Qinghai, Ningxia, Xinjiang | 16.67% | Neimeng, Liaoning, Heilongjiang, Zhejiang, Shandong, Hainan, Qinghai, Ningxia, Xinjiang | 30.00% |

(3) In terms of spatial layout, the comprehensive efficiency of the western regions greatly improved. Regional differentiation of comprehensive efficiency shifted from a pattern of east > central > west to west > central > east. The average efficiency values in 2011 were 0.413, 0.308, and 0.305. The average efficiency values in 2019 were 0.611, 0.626, and 0.628. The efficiency of eastern, central, and western provinces all increased, and the growth rate of the western provinces was significantly faster than that of the eastern and central regions. The comprehensive efficiency of tourism in coastal areas was generally higher than that of inland areas in 2011, while in 2019, many provinces in the central region improved their efficiency values, breaking the gap of huge coastal–inland differences and improving the efficiency value of the western regions.

*4.3. Empirical Results*

4.3.1. Spatial Correlation Analysis

Spatial correlation analysis is a necessary step to determine whether a research object has spatial correlation before a spatial economic model is used. Moran's I index is commonly used to test spatial correlation. In this paper, an adjacent spatial weight matrix was used to calculate the global Moran's I index for 30 Chinese provinces from 2011 to 2019. The results are shown in Table 3. Moran's I index of tourism efficiency and regional economic growth in China from 2011 to 2019 is significantly positive, indicating that there is positive spatial autocorrelation between tourism efficiency and regional economic growth in all provinces of China; a spillover effect exists. The regions with high tourism efficiency and regional economic growth will have positive impacts on surrounding areas.

**Table 3.** Moran's I index of tourism efficiency and economic growth from 2011 to 2019.

| Time | Tourism Efficiency | | | Economy Growth | | |
| --- | --- | --- | --- | --- | --- | --- |
| | **Moran's I** | **Z-Value** | ***p*-Value** | **Moran's I** | **Z-Value** | ***p*-Value** |
| 2011 | 0.149 | 1.546 | 0.022 | 0.422 | 3.718 | 0.000 |
| 2012 | 0.198 | 1.104 | 0.070 | 0.591 | 5.147 | 0.000 |
| 2013 | 0.182 | 1.957 | 0.038 | 0.598 | 5.201 | 0.000 |
| 2014 | 0.174 | 1.697 | 0.090 | 0.422 | 3.718 | 0.000 |
| 2015 | 0.145 | 1.469 | 0.042 | 0.585 | 5.079 | 0.000 |
| 2016 | 0.177 | 1.716 | 0.046 | 0.586 | 5.080 | 0.000 |
| 2017 | 0.185 | 1.718 | 0.023 | 0.591 | 5.110 | 0.000 |
| 2018 | 0.170 | 1.847 | 0.097 | 0.593 | 5.118 | 0.000 |
| 2019 | 1.141 | 1.820 | 0.012 | 0.592 | 5.097 | 0.000 |

4.3.2. Analysis of the Spatial Econometric Model

(1) Research on the whole space

The three spatial economic models, SAR, SEM, and SDM, were used to improve the authenticity of the empirical results. First, the Hausman test indicated that a random-effects model should be used in this study. Therefore, random-effects results for the SAR, SEM, and SDM models were calculated and are shown in Table 4.

Models I, III, and V in Table 4 show the independent effects of the internet index and tourism efficiency on economic growth. Due to the interactions between the two variables, "Internet plus Tourism", a cross item of the internet index and tourism efficiency, was introduced into the model to study its synergic effects on economic growth. The results are presented in Models I, III, and V in Table 4.

Table 4 demonstrates that the direction and significance of the three models are consistent except for the change of coefficient, which verifies the robustness of the estimation results to a certain extent. In Models I, III, and V, the coefficient of the internet index, tourism efficiency, transport infrastructure, and industrial structure all pass the 5% significance test. The degree of openness did not pass the significance test. From the perspective of R-squared, the spatial Durbin model fit better, so the following conclusions are drawn from that model.

The results show that the internet index and tourism efficiency have a significant positive effect on economic growth, which is consistent with the results of scholars such as Tang [41] and Li [42]. Therefore, H1 and H2 are proven correct. The result shows that developing the internet and tourism industries are the main means to improving economic growth at this stage. The rise of the Internet and the large-scale use of information technology not only promote technological progress and regional innovation by encouraging knowledge diffusion and improving total factor productivity [1,43], but they also accelerate the accumulation of human capital and improve labor productivity [44]. The internet also

can promote industrial upgrading [45] and the transfer of the labor force from primary industries to secondary and tertiary industries. Therefore, under the new normal, the internet has become a booster of economic growth.

**Table 4.** Spatial econometric model of the "Internet plus Tourism" index, tourism efficiency, and economic growth.

| Variable | SDM | | SEM | | SAR | |
|---|---|---|---|---|---|---|
| | Model I | Model II | Model III | Model IV | Model V | Model VI |
| Internet | 0.372 *** (4.49) | | 0.369 *** (3.40) | | 0.436 *** (4.75) | |
| Tourism | 0.118 * (0.82) | | 0.419 *** (3.56) | | 0.340 ** (2.30) | |
| Internet plus Tourism | | 0.446 *** (4.07) | | 0.260 *** (3.81) | | 0.495 *** (4.68) |
| Transportation | 0.052 ** (2.00) | 0.075 *** (2.82) | 0.043 * (1.65) | 0.054 ** (2.05) | 0.095 *** (1.73) | 0.143 *** (5.55) |
| Industry | −0.380 *** (−6.19) | −0.315 *** (−5.00) | −0.396 *** (−6.50) | −0.374 *** (−6.05) | −0.311 *** (−5.08) | −0.082 (−1.49) |
| Opening | 0.333 * (1.78) | 0.551 *** (2.88) | 0.352 * (1.93) | 0.480 *** (2.64) | −0.036 (−0.20) | 0.083 (0.42) |
| Cons | 20.18 *** (6.72) | 14.99 *** (5.83) | 102.9 *** (50.90) | 109.0 *** (115.05) | 12.35 *** (5.55) | 7.438 *** (4.40) |
| $\rho/\lambda$ | 0.627 *** (13.20) | 0.820 *** (29.50) | 0.969 *** (153.39) | 0.971 *** (169.19) | 0.772 *** (24.56) | 0.919 *** (53.61) |
| $R^2$ | 0.9840 | 0.9662 | 0.8302 | 0.4899 | 0.9782 | 0.9411 |

* $p < 0.05$, ** $p < 0.01$, *** $p < 0.001$.

Secondly, "Internet plus Tourism", the cross term of the internet index and tourism efficiency, has a significantly positive effect on economic growth that is significantly greater than the internet index and tourism efficiency independently, indicating that "Internet plus Tourism" has a "1 + 1 > 2" effect, thus confirming H3. At present, the deep integration of China's tourism industry and the internet has improved resource integration within the tourism industry, formed a new mode of industry development, and effectively realized the development potential and dynamic vitality of the tourism industry in a way that "drives economic growth through tourism". Tourists have undergone a transformation process from passive receivers of tourism products to co-creators and leaders of the tourism industry chain [46]. This has encouraged tourists to purchase more tourism products, greatly promoting the dynamic demand and universality of tourists and boosting local tourism.

(2) Research on regional space

The study analyzed regional differences across the three regions of east, central, and west. A spatial Durbin model was then used to empirically analyze the spillover effect of each variable. The results are listed in Table 5.

The coefficients of the internet index, tourism efficiency, and "Internet plus Tourism" in the eastern region were 0.452, 0.326, and 0.535, respectively, and the spillover effects of the three are significant and have strong correlation driving effects. Therefore, a vertical comparison denotes that the eastern region's economic growth is a typical "Internet plus Tourism"-oriented mode. The coefficients of the internet index, tourism efficiency, and "Internet plus Tourism" in the central region were 0.307, −0.216, and 0.0407, respectively. It indicates that tourism development has brought negative spillover effects and is still at a relatively low level in central China. The economic growth of the central region fits an internet-oriented development model. The coefficients of the internet index, tourism efficiency, and "Internet plus Tourism" in western China were 0.193, −0.013, and 0.258, respectively. Tourism development had negative effects. This is related to the economic

development, market relevance, innovation stock, and innovation potential of the western region.

**Table 5.** Regional spatial econometric models of the internet, tourism efficiency, and economic growth.

| Variable | East | | Central | | West | |
|---|---|---|---|---|---|---|
| | Model I | Model II | Model III | Model IV | Model V | Model VI |
| Internet | 0.452 *** (3.93) | | 0.307 *** (3.65) | | 0.193 * (0.67) | |
| Tourism | 0.326 ** (2.50) | | −0.216 (−1.18) | | −0.0130 * (−0.05) | |
| Internet plus Tourism | | 0. 535 *** (2.64) | | 0.407 (0.35) | | 0.258 (1.17) |
| Transportation | −0.053 (−0.76) | 0.076 (1.16) | 0.0498 (1.33) | 0.108 ** (2.53) | 0.0931 ** (2.04) | 0.0998 ** (2.23) |
| Industry | −0.418 *** (−3.33) | −0.469 *** (−4.00) | −0.238 *** (−2.79) | −0.196 ** (−1.97) | −0.109 *** (−0.99) | −0.085 (−0.84) |
| Opening | 0.357 (0.20) | 0. 666 (0.91) | −0.293 (−1.64) | −0.237 ** (−2.28) | 0.296 (0.90) | 0.528 (0.68 |
| Cons | 25.08 *** (4.61) | 27.69 *** (5. 32) | 27.00 *** (5.37) | 27.21 *** (5.35) | 23.41 *** (4.55) | 18.42 *** (4.56) |
| $\rho / \lambda$ | 0.583 *** (8.12) | 0.727 *** (13.60) | 0.608 *** (9.17) | 0.971 *** (169.19) | 0.581 *** (6.84) | 0.796 *** (17.37) |
| $R^2$ | 0.9743 | 0.9553 | 0.9878 | 0.4899 | 0.9810 | 0.9394 |

* $p < 0.05$, ** $p < 0.01$, *** $p < 0.001$.

Comparative analysis of the effects of the variables on economic growth helps formulate targeted policy recommendations to promote economic growth and regional integration development. The results demonstrate that the economic effects of the internet index, tourism efficiency, and "Internet plus Tourism" in the eastern region are more prominent. The midwest region needs to realize that "Internet plus Tourism" is an important variable for catching up with the eastern region and achieving economic take-off. Therefore, government departments and social forces should pay attention to the role and value of the internet in the process of tourism development so as to increase investment in key provinces in the central and western regions.

## 5. Conclusions and Suggestions

### 5.1. Main Conclusions

Using various provinces' internet and tourism development data from 2011 to 2019, this paper measures the efficiency of tourism development in various provinces based on DEA and puts forward an internet index to reflect the current development of regional internet services. Empirical research on the relationship between the internet, tourism development, and economic growth through a spatial measurement model shows that:

(1) During the study period, the internet index of each province was significantly different and the index points shifted with time. Regarding hierarchical structure, the provinces with significant changes were concentrated in the second and third levels. For example, the growth of Shanxi, Jilin, Guizhou, and Guangxi had significant advantages, which provided the core thrust for the internet's high-speed development. Regarding spatial distribution, the plate pattern is higher in the coastal areas and lower in the central and western inland areas.

(2) In terms of tourism efficiency, comprehensive efficiency is on the rise, and the average efficiency value rose from 0.346 in 2011 to 0.621 in 2019. The internal hierarchical structure is polarized, with the number of high and low levels increasing and the number

of intermediate levels decreasing. In terms of spatial layout, the comprehensive efficiency of the central region improved and an S-shaped pattern was formed.

(3) The internet index and tourism efficiency had significant spatial effects on the process of economic growth. There is a close interactive relationship between the internet index and tourism efficiency. Not only did the independent values of the internet index and tourism efficiency have positive effects on economic growth, but the combined value of "Internet plus Tourism" had a positive effect that was significantly greater. The study shows that "Internet plus Tourism" has a "1 + 1 > 2" effect on economic growth.

(4) From the regression results of the subregions, it can be seen that the internet index in the eastern, central, and western regions is conducive to economic growth. For tourism efficiency, only the eastern region saw a positive economic effect, while the central and western regions had negative to varying degrees. However, "Internet plus Tourism" had a positive impact on economic growth in different regions, which indicates that China should focus on increasing internet and tourism infrastructure in the midwest and promote local economic development through the integration of the internet and tourism.

### 5.2. Policy Recommendations

The rapid development of tourism and its strong momentum and extensive industrial scale have gradually forged it into an indispensable industry for global economic development that plays an important role in promoting regional economies, social employment, and regional cultural exchanges and development. The development of "Internet plus Tourism" grows with each passing day and has come to play an increasingly important role in the development of the national economy. At present, China is still in the "primary stage" of development, but its potential and space should not be underestimated. Therefore, China's "Internet plus Tourism" mode should be further emphasized from national and regional perspectives in order to make full use of the internet's application pace, big data, and convenient features and to accelerate the integration of "Internet plus Tourism".

(1) Strengthen the construction of both the internet and tourism infrastructure in all regions and, especially, increase investment in the central and western regions.

Infrastructure provides a vital foothold for both internet and tourism development. As the leading factor for successful internet and tourism development, increased infrastructure will create more development space for the industry. Therefore, the government should increase the investment in infrastructure in these two areas, especially in the slow-developing central and western regions, so as to promote the growth of regional economies.

(2) Strengthen the integration of the internet and the tourism industry and increase the penetration rate of the internet in the tourism industry.

Due to spatial constrictions, the traditional tourism industry cannot quickly and accurately transmit information to consumers, resulting in poor consumer experiences. The internet can effectively use the tourism information provided by big data, analyze the real needs of consumers, provide consumers with accurate and personalized tourism programs, greatly improve consumer satisfaction, and integrate more high-quality tourism resources to improve the economic benefits of local tourism. At present, China's tourism industry has not been fully integrated with other countries, and there is still much room for improvement. Therefore, we must apply constant innovation. The application of internet technologies, such as the Internet of Things, cloud computing, big data, and artificial intelligence in the tourism industry can greatly accelerate the integration speed of the two and improve the growth of regional economies.

(3) Establish a perfect development mode that centers "online coordinate offline" and "offline sublimate online".

Fundamentally, tourism is governed by the experience economy and the personalized economy. Internet technology is expected to offer personalized recommendations and solutions by accurately calculating and predicting tourists' needs, which greatly reduces consumer costs and increases travel efficiency. Therefore, the government should encourage and actively promote "Internet plus Tourism" and employ a development mode that blends

"online coordinate offline" and "offline sublimate online". That is to say, the government should improve the integration of "online coordinate offline" via strengthened consumer channel integration and use intelligent applications (such as AI) to boost "offline sublimate online" and ensure the offline needs and activities of consumers are spread more accurately through the internet.

## 6. Limitations and Future Work

This research was subject to some limitations, which should be considered for further research. First, the measurement index of "Internet plus" we selected was relatively simple and may cause some deviations in the overall research results. Therefore, it is necessary to further accurately measure "Internet plus" in the future. Second, when studying the impact of the cross-integration of "Internet plus" and tourism on economic growth, we only multiplied the two indicators and did not establish an effective indicator system to measure it. The next step should establish a specific indicator system. Third, another limitation was the choice of samples and locations. We studied the relationship from the perspective of the whole and regions. The next step can be to analyze the relationship from the perspective of provinces.

**Author Contributions:** Conceptualization, R.D.; methodology, M.H.; software, M.H.; validation, R.D. and M.H.; formal analysis, M.H.; investigation, M.H.; resources, M.H.; data curation, M.H.; writing—original draft preparation, M.H.; writing—review and editing, R.D.; visualization, M.H.; supervision, R.D.; project administration, R.D.; funding acquisition, R.D. All authors have read and agreed to the published version of the manuscript.

**Funding:** This work was supported by the China National Key R&D Program during the 13th Five-year Plan Period (2016YFC0801906).

**Institutional Review Board Statement:** Not applicable.

**Informed Consent Statement:** Informed consent was obtained from all subjects involved in the study.

**Data Availability Statement:** All data used in this evaluation are available publicly and also from the authors. Please contact author Meng Huang with data requests (dingrijia@cumtb.edu.cn).

**Conflicts of Interest:** The authors declare no conflict of interest.

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
