# Peer review of "The Spatial Difference of “Internet plus Tourism” in Promoting Economic Growth"

_sustainability, doi:10.3390/su132111788_

Round 1

Reviewer 1 Report

The current manuscript is written and presented with details in the research steps and results. Some minor points are required to improve or clarify:

  1. How can this research topic be framed in the field of sustainability?
  2. Explain why some of the variables are logarithmic.
  3. There are no references to the Economies journal. This makes me think that the paper is not necessarily related to the scope of the journal.

Reviewer 2 Report

In my opinion, the topic is very current and it is a point at issue. The validity of the problematic is especially important within COVID times. My suggestion is that the hypothesis should be more specific/determinated because they are a little too obvious (because it is commonly known that tourism influences economic growth on every level of country's development).

Reviewer 3 Report

The investigated topic is of great interest to readers. The research is well organized, and it is exposed in the document in a precise way: the issue is contextualized, the gap is identified, the objective is set, the statistical tools to achieve it are decided, the results are presented and a great strength, are the practical implications exposed by the authors.
My recommendations for improving the document are:
1. In the abstract, improve the exposure of the objective, it is not at all clear. It is necessary to read the entire document to observe the stated objective.
2.- In the abstract, expose in more detail the methodology, which models are used to obtain the results and, therefore, the conclusions and recommendations that the authors provide.
3.- In the introductory section: the topic is appropriately contextualized, it is identified what other researchers have done and the gap that gives rise to the objective of the research is observed. However, it would be necessary to include in the introduction the objective clearly, the methodology briefly and a final paragraph in which it is explained in which sections the document is divided.
4.- The legend of figures 2 and 3 is not read. It is important to improve the display of the maps. It would be necessary to include a map with the name of the provinces, in this way the reader can see the maps (results) more visually.
5.- Include the limitations of the investigation.
6.- See the authors guide, both the in-text citations and the citations section do not conform to the rules for authors.
I congratulate the authors for their research and encourage them to make the changes and suggestions made.
